# Comparison of the internal fit of metal crowns fabricated by traditional casting, computer numerical control milling, and three-dimensional printing

**Wei-Ting Chou**[1,2], **Chuan-Chung Chuang**[1,2], **Yi-Bing Wang**[1], **Hsien-Chung Chiu**[1,2]*

**1** Division of Prosthodontics, Tri-Service General Hospital, Taipei, Taiwan, R.O.C, **2** School of Dentistry, National Defense Medical Center, Taipei, Taiwan, R.O.C

* chiud1121@yahoo.com.tw

**Data Availability Statement:** All data generated or analysed during this study are included in this published article and its supplementary information files.

## Abstract

This experimental study aimed to compare the internal fit (marginal fit and internal discrepancy) of metal crowns fabricated by traditional casting and digital methods (computer numerically controlled (CNC) milling and three-dimensional [3D] printing). Thirty standard master abutment models were fabricated using a 3D printing technique with digital software. Metal crowns were fabricated by traditional casting, CNC milling, and 3D printing. The silicon replica method was used to measure the marginal and internal fit. A thin layer of low-viscosity polyvinyl siloxane material was placed inside each crown and on the die (like a seat) until the material was set. Replicas were examined at four reference points under a microscope: the central pit (M1), cusp tip (M2), axial wall (M3), and margin (M4). The measured data were analyzed using a one-way analysis of variance (ANOVA) to verify statistical significance, which was set at $p < 0.05$. In the traditional casting group, the minimum distance measured was at M3 ($90.68 \pm 14.4$ μm) and the maximum distance measured was at M1 ($145.12 \pm 22$ μm). In the milling group, the minimum distance measured was at M3 ($71.85 \pm 23.69$ μm) and the maximum distance measured was at M1 ($108.68 \pm 10.52$ μm). In the 3D printing group, the minimum distance measured was at M3 ($100.59 \pm 9.26$ μm) and the maximum distance measured was at M1 ($122.33 \pm 7.66$ μm). The mean discrepancy for the traditional casting, CNC milling, and 3D printing groups was 120.20, 92.15, and 111.85 μm, respectively, showing significant differences ($P < 0.05$). All three methods of metal crown fabrication, that is, traditional casting, CNC milling, and 3D printing, had values within the clinically acceptable range. The marginal and internal fit of the crown was far superior in the CNC milling method.

## Introduction

With the advent of the digital age, dentistry has become increasingly digitalized, combining computer-aided design/manufacturing software (CAD/CAM) and three-dimensional (3D) printing [1]. Dr. Francois Duret, who performed digital scans or indirect model scans in

**Funding:** The author(s) received no specific funding for this work.

**Competing interests:** The authors have declared that no competing interests exist.

patients as early as 1955, was the first to translate the concept of CAD/CAM from an industrial technique to the field of dentistry. In 1971, the CAD/CAM technology was officially introduced for dental applications. In 1985, Dr. Werner Mormann became the second developer of CAD/CAM in dentistry, designing an intraoral camera and a milling machine to fabricate dentures in the dental office. Dr. Matts Anderson followed as the third developer and used it to fabricate metal restorations [2].

In simple terms, the process can be described as follows: the dentist creates a digital dataset on the computer (computer-aided design, CAD) and then designs a 3D object; the data of the 3D object are then transferred to the milling machine or a 3D printer, which creates a physical object from these data.

Dental CAD/CAM systems involve using new materials, cost control, quality control, and reduced time to fabricate dental crowns in the clinic [3]. Fabricating dental crowns is a daily task for dentists. The key to a successful dental crown is the fit between the margin and the crown's interior. A crown that is not a close fit can cause dental problems. The clinically acceptable crown gap is 120 μm [4], and when this gap widens, dental problems such as secondary tooth decay, plaque accumulation, and periodontal disease are likely to occur [5].

The marginal gap values of various materials, such as zirconia, lithium disilicate, polymethyl methacrylate (a temporary crown material), and traditional casting metal crowns, are approximately 56 [6], 74 [7], 161 [8,9], and 88 μm [10], respectively, which meet clinical standards based on using the dental CAD/CAM system. This experimental study aimed to compare the marginal and internal fit of metallic cobalt and chromium alloys between traditional manually cast crowns and digitally processed crowns (CNC-milled and 3D printed crowns).

The null hypothesis is that the internal fit of digitally fabricated crowns is better than those of traditionally fabricated crowns.

## Materials and methods

### Preparation of the abutment tooth design

The experimental model was designed using two types of software. First, the Exocad DentalCAD (exocad GmbH) design software was used to select the mandibular first molar model from the tooth library, and the reduced pontic tool was used to cut the entire crown back into the shape of the abutment. Next, the Meshmixer (Autodesk, San Rafael, CA, USA) 3D modeling software was used to design the base and convert it into a STereoLithograph (STL) format.

### Resin abutment fabrication

The STL file of the completed design was printed using a Stratasys Objet260 Connex 3D printer. VeroWhite resin was used as the material. The weight of the resin abutment was 14 g, the thickness of the printed layer was 0.03 mm, and a total of 30 pieces were printed (Fig 1).

### Scanning and crown fabrication

A traditional wax pattern was applied to the resin abutment to complete the wax pattern of the crown. The wax pattern was scanned (Ceramill Map 300; AmannGirrbach AG), and the morphology of the crown was completely replicated using a computer (Fig 2). Three different fabrication processes were used, with cobalt-chromium alloy as the material. Ten groups were fabricated by wax printing and traditional casting, 10 using the metal printer (ConceptLaser, Germany), and 10 by CNC milling using the Ceramill Sintron (Amann Girrbach AG) (Fig 3).

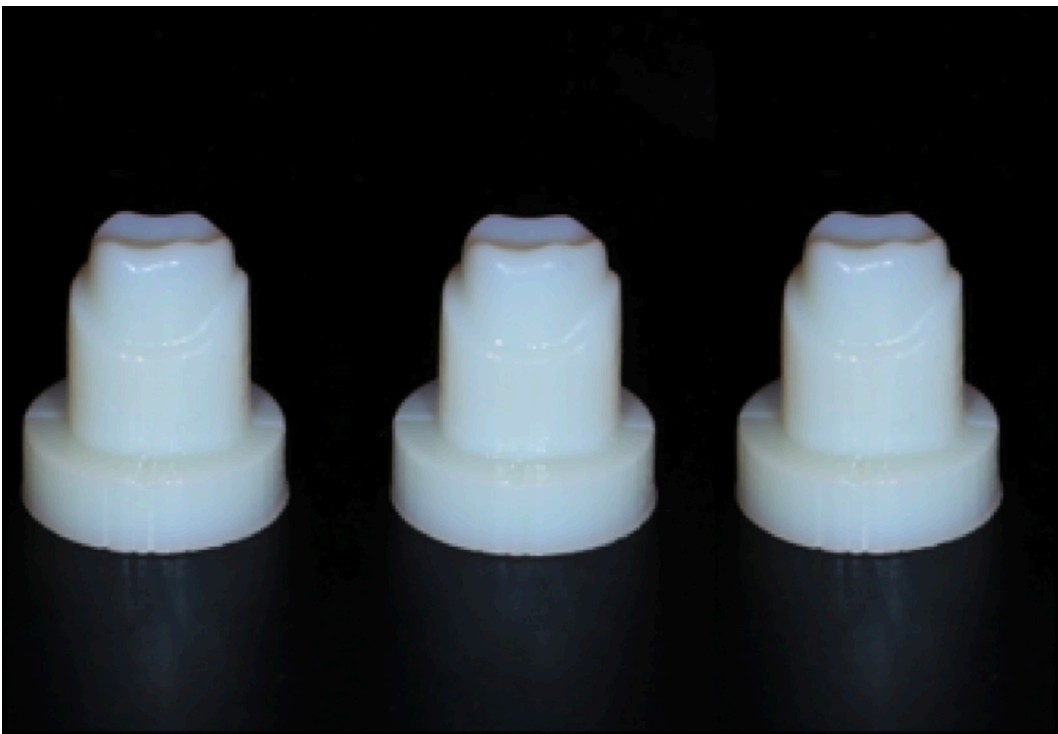

**Fig 1. Three-dimensional printed standardized resin abutment models.**

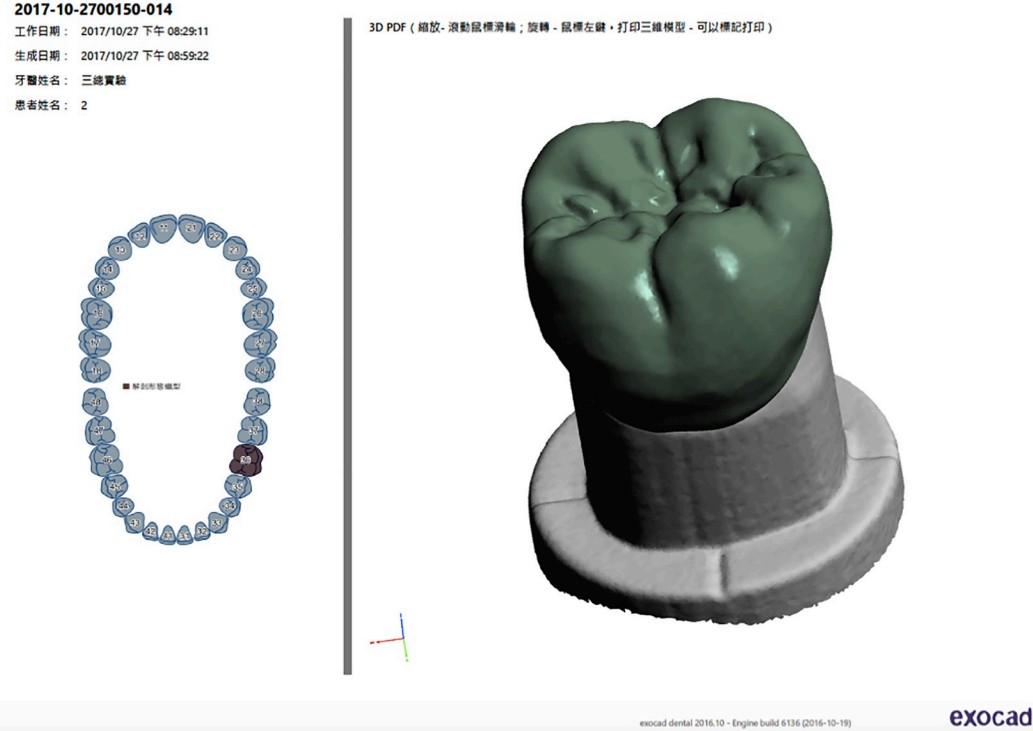

**Fig 2. Computerized copy of the wax pattern.**

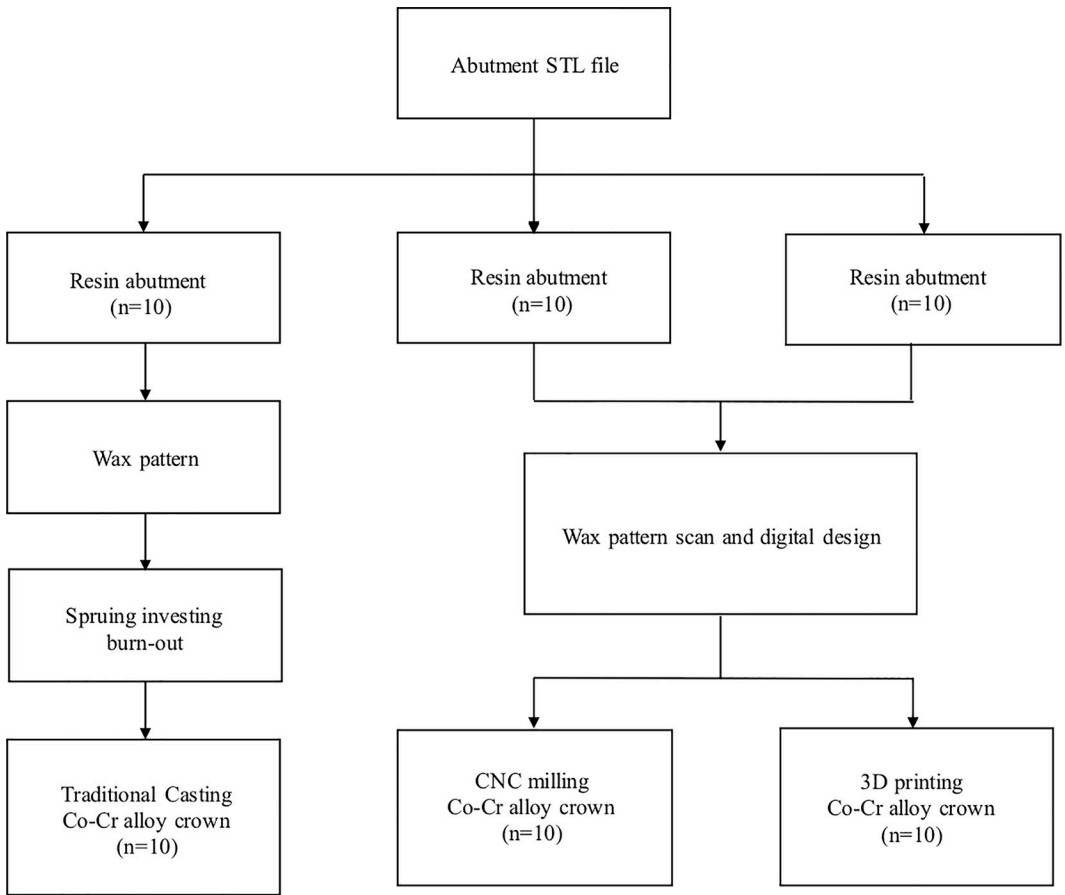

**Fig 3. Flowchart of the crown fabrication process.**

### Silicone replica technique

The silicone replica technique (Fig 4) [11,12] was used to observe the internal fit and margin of the crown. Thirty sets of different crowns were filled with light body silicone (Take 1 Advanced light body wash; Kerr) and pressed into the resin abutment with a constant force of 50 N. At this time, a thin layer of silicone impression material was observed, which revealed a gap between the interior of the crown and the resin abutment. When the material was completely hardened, the crown and resin abutment were separated, and medium-body silicone (Take 1 Advanced regular body wash; Kerr) was injected into the crown. After completion, the reprinted silicone was cut in half from the center to the mesiodistal direction.

### Measurements

The orange silicone layer represents the gap between the model and crown (Fig 4) [13]. The four measurement points were the central pit (M1), cusp tip (M2), axial wall (M3), and margin (M4) (Fig 5). A dental microscope (Zumax, Jiangsu, China) was used at a magnification of 23X, and a Sony a7 single-lens digital camera was used for imaging. The gap from M1 indicates the position of the occlusal center of the model and the interior of the crown, the gap from M2 the position between the model and cusp tip in the interior of the crown, the gap from M3 the distance between the middle axial wall of the model and the inside of the crown, and the gap from M4 the marginal gap between the crown and model. ImageJ software was used for the linear measurements.

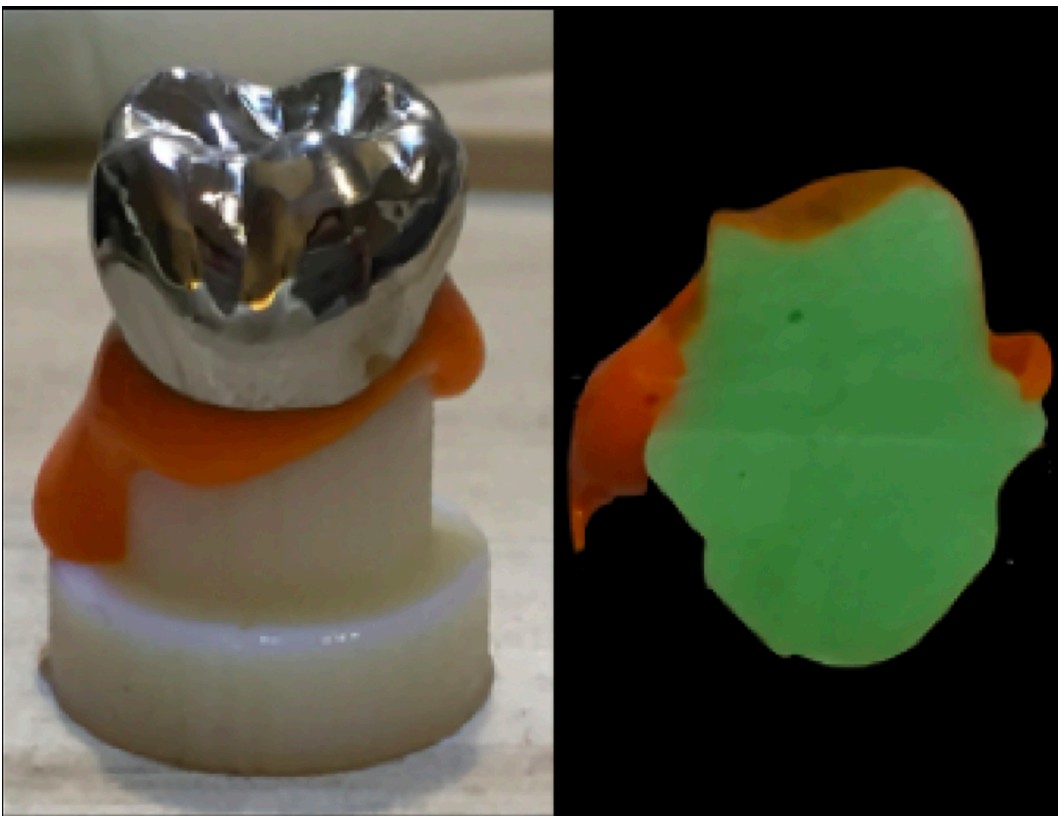

**Fig 4. Silicone replica technique.**

### Statistical analysis

Statistical analysis was performed using the Statistical Package for Social Sciences statistical software (version 22.0.0; IBM Corp., Armonk, NY, USA). The statistical method used was the analysis of variance (ANOVA). The items calculated included statistics for the four measurement sites, M1–M4. The significance level was set at $P < 0.05$.

## Results

The results for the four measurement sites were obtained for all three groups (S1 File). In the traditional casting group, the minimum measured distance was at M3 ($90.68 \pm 14.4$ μm), and the maximum measured distance was at M1 ($145.12 \pm 22$ μm). In the CNC milling group, the minimum measured distance was at M3 ($71.85 \pm 23.69$ μm), and the maximum measured distance was at M1 ($108.68 \pm 10.52$ μm). In the 3D printing group, the minimum measured distance was at M3 ($100.59 \pm 9.26$ μm), and the maximum measured distance was at M1 ($122.33 \pm 7.66$ μm) (Table 1, Fig 6). The mean discrepancy measurements of the traditional casting, CNC milling, and 3D printing groups were 120.20, 92.15, and 111.85 μm, respectively (Fig 7). The differences in fit among the three methods were statistically significant.

## Discussion

The digitalization of dentistry has many advantages. CAD/CAM processing can reduce unnecessary wasting of materials and fabrication time and cost [3]. Since 2011, 3D printing has enjoyed a worldwide wave of popularity, especially selective laser melting metal printing

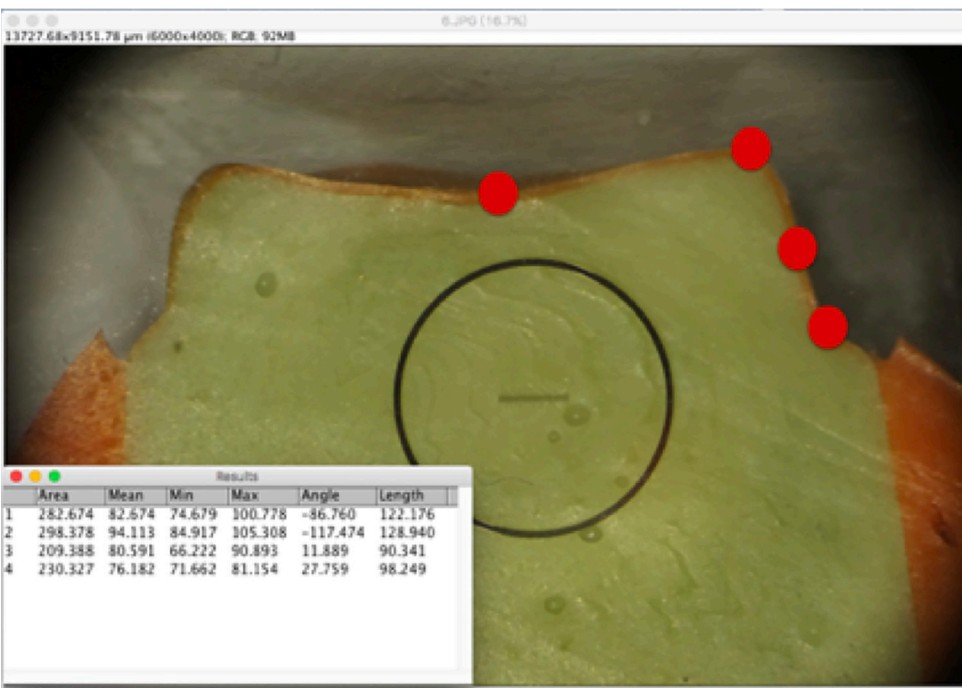

**Fig 5. Measurement sites, M1 (central pit), M2 (cusp tip), M3 (axial wall), and M4 (margin).**

technology [14,15]. With the rapid development and integration of software and hardware technologies, CAD/CAM design and 3D printing technology have been used to fabricate high-quality crowns with greater accuracy in addition to facilitating crown design by clinicians.

The fit between internal crowns is key to clinical success [16]. The internal and marginal gaps were defined as the vertical distances between the crown and tooth abutment. Large marginal gaps can result in cement hydrolysis, which leads to dental problems such as dental plaque accumulation, gingival inflammation, secondary caries, and discoloration of the gingival margin [14,17].

Clinically, some cement space must be left between the crown and the abutment. Fransson et al. indicated that this space should be between 20 and 40 μm [18], whereas Beuer et al. indicated that a gap of 50 μm yields the most suitable crown fit [19].

Bhaskaran et al. indicated that the clinically acceptable crown marginal fit is between 10 and 160 μm, and the internal crown fit is between 81 and 136 μm [20]. In recent years, zirconia crowns have been the most commonly used for CAD/CAM processing [16]. Reich et al. (2005)

**Table 1. Mean and standard deviation of the difference in each group at each measurement site (units: μm).**

| Measurement site | Metal crown fabrication method | | | P-value |
|---|---|---|---|---|
| | Traditional casting | CNC milling | 3D printing | |
| | Mean ± SD | Mean ± SD | Mean ± SD | |
| Central pit | 145.16 ± 22 | 108.68 ± 10.52 | 122.33 ± 7.66 | < 0.001 |
| Cusp tip | 123.8 ± 13.35 | 94.31 ± 10.26 | 113.19 ± 9.9 | < 0.001 |
| Axial wall | 90.68 ± 14.4 | 71.85 ± 23.69 | 100.59 ± 9.26 | 0.003 |
| Margin | 121.18 ± 16.25 | 93.79 ± 10.47 | 111.3 ± 12.3 | < 0.001 |

CNC, computer numeric control; 3D, three-dimensional; M1, central pit; M2, cusp tip; M3, axial wall; M4, margin.

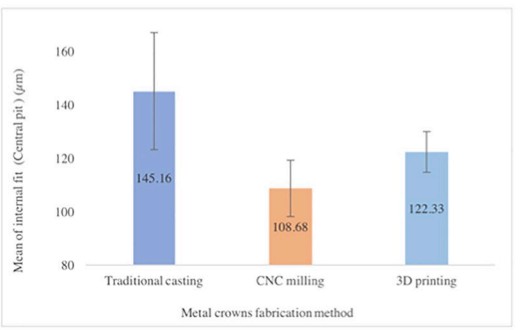
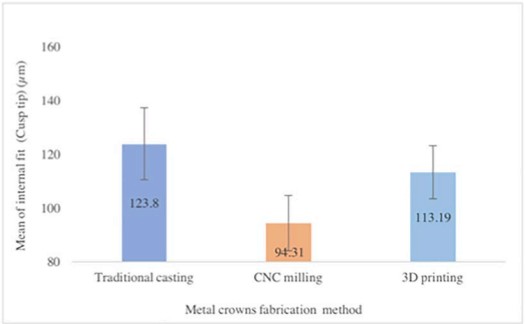
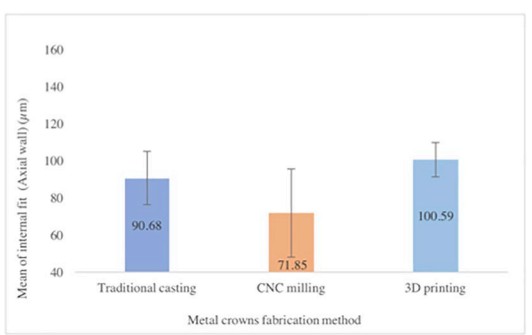
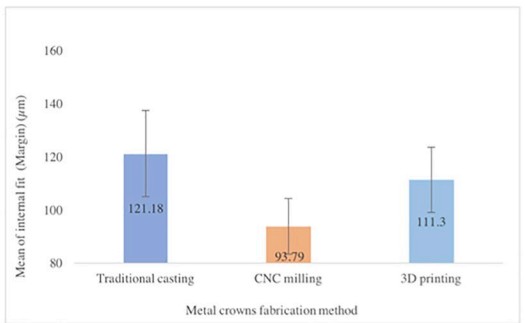

**Fig 6. Average values at the measurement sites.**

used zirconia crowns to study the internal fit and obtained results at the four measurement sites for all three groups, the median marginal gap was 75 μm. For the traditional casting crown is 54 μm [17]. These results are within the research range and are clinically acceptable for dentists and patients.

The progress of metal crown printing has gradually stabilized in recent years. The use of titanium alloys, stainless steel, and other materials for medical applications is currently sophisticated. However, some medical materials cannot be used directly in clinics because of existing

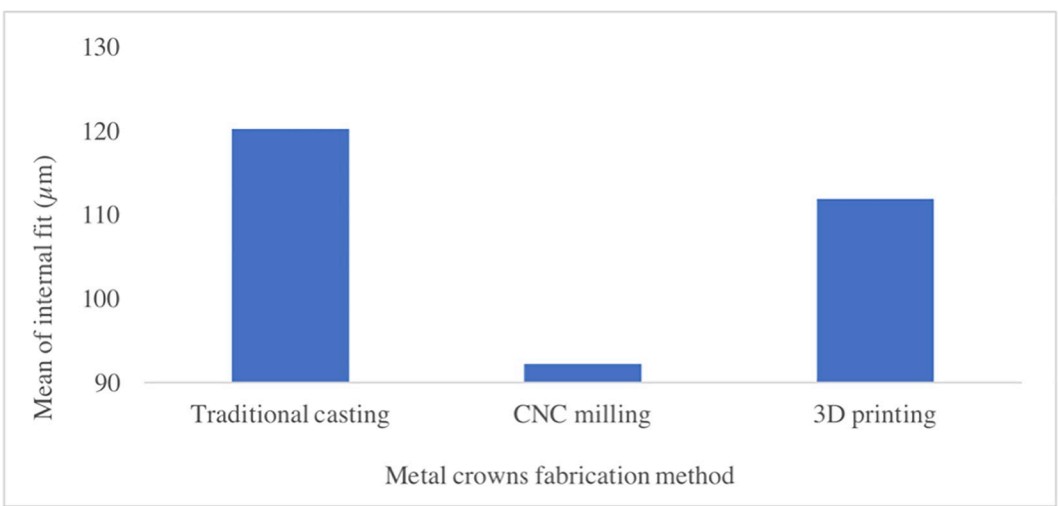

**Fig 7. Average values for each group.**

regulations. Dental-related products have lesser restrictions, and some products are already available in the market. For example, the cobalt-chromium alloy powder used in this study was directly fabricated by metal printing to produce the finished product [21].

In the traditional casting method, the molten alloy used for crown restoration shrinks upon solidification, and cobalt-chromium alloys shrink by as much as 2%. If the mold is not made correspondingly larger than the original wax pattern, the resultant cast will be much smaller. Therefore, for casting crowns, it is necessary to compensate for the solidification-related shrinkage of the alloy used by expanding the mold size to at least equal the shrinkage size. One of the mechanisms to produce mold expansion is wax pattern expansion; therefore, proper control by technicians may also influence the crown fit result.

Over the years, the CAD/CAM fabrication method has gained popularity over the conventional method in dentistry, and dentists are moving ahead with the trend by taking advantage of the opportunities provided by the CAD/CAM technology to equip their clinics or laboratories with the latest materials that will make work easier and more efficient. One of the benefits of dental CAD/CAM technology is that it is time-saving, enabling faster dental crown production. It reduces the time to produce crowns to 2–5 days instead of 2–3 weeks required with the conventional procedure. With the CAD/CAM technology in dentistry, the outcome of restorations can be predicted. Devoid of traditional impressions requiring several processes, intraoral-extraoral scanning gives patients a more convenient dental restoration experience in the clinic.

In this study, the internal fit of crowns fabricated using different fabrication processes was determined. Traditional casting, CNC milling, and 3D printing techniques were used, and the results showed that the mean discrepancy measurements for traditional casting, CNC milling, and 3D printing were 120.20, 92.15, and 111.85 μm, respectively. The results for all groups were within the clinically acceptable range of 120 μm proposed by McLean [4]. The results of this study indicated that the performance of digitally fabricated crowns (casting, CNC milling, and 3D printing) was better than that of traditionally fabricated crowns.

The limitations of this study include the *in vitro* setting, which might not reflect real conditions in patients' mouths and daily clinical practice. In addition, even though the CAM equipment of the CAD/CAM system, which included the milling machine and 3D printer, has extremely outstanding surface illumination and high precision, more specimens and various milling machines and printers were not used. Therefore, there is a need to secure a greater number of specimens and additional equipment in future studies, and the use of human teeth would be ideal for simulating a clinical procedure.

## Conclusions

This study showed that the fit of metal crowns fabricated by traditional casting, CNC milling, and 3D printing all met clinical requirements (120 μm). The CAD/CAM fabrication methods performed better than traditional fabrication methods. In particular, metal crowns processed by CNC milling exhibited the best performance as regards internal and marginal fit. However, more samples are required for future research and evaluation.

## Supporting information

**S1 File. Crown measurement data.**
(XLSX)

## Acknowledgments

The authors acknowledge the services provided by the staff of the Division of Prosthodontics, Tri-Service General Hospital, National Defense Medical Center.

## Author Contributions

**Conceptualization:** Wei-Ting Chou.

**Data curation:** Wei-Ting Chou.

**Formal analysis:** Wei-Ting Chou.

**Investigation:** Wei-Ting Chou.

**Methodology:** Wei-Ting Chou.

**Project administration:** Wei-Ting Chou.

**Resources:** Wei-Ting Chou.

**Software:** Wei-Ting Chou.

**Supervision:** Chuan-Chung Chuang, Yi-Bing Wang, Hsien-Chung Chiu.

**Validation:** Wei-Ting Chou.

**Visualization:** Wei-Ting Chou.

**Writing – original draft:** Wei-Ting Chou.

**Writing – review & editing:** Chuan-Chung Chuang, Yi-Bing Wang, Hsien-Chung Chiu.

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
