## [Editor Report · Decision Letter 0]

21 Jun 2021

PONE-D-21-12569

Comparison of the internal fit of metal crowns fabricated by traditional casting, computer numerical control milling, and three-dimensional printing

PLOS ONE

Dear Dr. Chou,

Thank you for submitting your manuscript to PLOS ONE. After careful consideration, we feel that it has merit but does not fully meet PLOS ONE’s publication criteria as it currently stands. Therefore, we invite you to submit a revised version of the manuscript that addresses the points raised during the review process.

Please see comments from the editor. Please address all comments and revise your manuscript accordingly.

We look forward to receiving your revised manuscript.

Kind regards,

Sompop Bencharit, DDS, MS, PhD, FACP

Academic Editor

PLOS ONE

Journal Requirements:

2. TC2 Complete.

Additional Editor Comments (if provided):

While the work is interesting and has merit for future publication, there are some clarifications and additional information needed. Please see below:

1) Please add a summary of technology used to mill and print metal crowns in the introduction, perhaps in the third paragraph.

2) Please add explicit hypothesis and what you have expected in the last part of the introduction.

3) The Discussion needs to explain the limitations of the study and how these limitations may be addressed in future experiments. This should be in the last part of the Discussion. Errors of the lost wax technique, scanning errors, printing errors, and measurement errors should also be discussed.

4) Please discuss clinical relevances and how the experiment may or may not applicable to clinic patients.
---

## [Author Response · Author response to Decision Letter 0]

16 Aug 2021

Response to Reviewers

1) Please add a summary of technology used to mill and print metal crowns in the introduction, perhaps in the third paragraph.

Response: In simple terms, the process can be described as follows: the dentist creates a digital dataset on the computer (computer-aided design, CAD) and then designs a 3D object; the data of the 3D object are then transferred to the milling machine or a 3D printer, which creates a physical object from these data. 

2) Please add explicit hypothesis and what you have expected in the last part of the introduction.

Response: The null hypothesis is that the internal fit of digitally fabricated crowns is better than those of traditionally fabricated crowns.

3) The Discussion needs to explain the limitations of the study and how these limitations may be addressed in future experiments. This should be in the last part of the Discussion. Errors of the lost wax technique, scanning errors, printing errors, and measurement errors should also be discussed.

Response: The limitations of this study include the in vitro setting, which might not reflect real conditions in patients’ mouths and daily clinical practice. In addition, even though the CAM equipment of the CAD/CAM system, which included the milling machine and 3D printer, has extremely outstanding surface illumination and high precision, more specimens and various milling machines and printers were not used. Therefore, there is a need to secure a greater number of specimens and additional equipment in future studies, and the use of human teeth would be ideal for simulating a clinical procedure. 

4) Please discuss clinical relevances and how the experiment may or may not applicable to clinic patients.

Response: In the traditional casting method, the molten alloy used for crown restoration shrinks upon solidification, and cobalt-chromium alloys shrink by as much as 2%. If the mold is not made correspondingly larger than the original wax pattern, the resultant cast will be much smaller. Therefore, for casting crowns, it is necessary to compensate for the solidification-related shrinkage of the alloy used by expanding the mold size to at least equal the shrinkage size. One of the mechanisms to produce mold expansion is wax pattern expansion; therefore, proper control by technicians may also influence the crown fit result.

Over the years, the CAD/CAM fabrication method has gained popularity over the conventional method in dentistry, and dentists are moving ahead with the trend by taking advantage of the opportunities provided by the CAD/CAM technology to equip their clinics or laboratories with the latest materials that will make work easier and more efficient. One of the benefits of dental CAD/CAM technology is that it is time-saving, enabling faster dental crown production. It reduces the time to produce crowns to 2-5 days instead of 2-3 weeks required with the conventional procedure. With the CAD/CAM technology in dentistry, the outcome of restorations can be predicted. Devoid of traditional impressions requiring several processes, intraoral-extraoral scanning gives patients a more convenient dental restoration experience in the clinic.

---

## [Editor Report · Decision Letter 1]

25 Aug 2021

Comparison of the internal fit of metal crowns fabricated by traditional casting, computer numerical control milling, and three-dimensional printing

PONE-D-21-12569R1

Dear Dr. Chiu,

We’re pleased to inform you that your manuscript has been judged scientifically suitable for publication and will be formally accepted for publication once it meets all outstanding technical requirements.

Kind regards,

Sompop Bencharit, DDS, MS, PhD, FACP

Academic Editor

PLOS ONE

Additional Editor Comments (optional):

Thank you for the revision and responses to the comments.
---

## [Editor Report · Acceptance letter]

3 Sep 2021

PONE-D-21-12569R1 

Comparison of the internal fit of metal crowns fabricated by traditional casting, computer numerical control milling, and three-dimensional printing 

Dear Dr. Chiu:

I'm pleased to inform you that your manuscript has been deemed suitable for publication in PLOS ONE. Congratulations! Your manuscript is now with our production department. 

Kind regards, 

on behalf of

Dr. Sompop Bencharit 

Academic Editor

PLOS ONE